# Human Development Index Is Associated with COVID-19 Case Fatality Rate in Brazil: An Ecological Study

**DOI:** 10.3390/ijerph19095306

**Published:** 2022-04-27

**Authors:** Camila Vantini Capasso Palamim, Matheus Negri Boschiero, Felipe Eduardo Valencise, Fernando Augusto Lima Marson

**Affiliations:** 1Laboratory of Cell and Molecular Tumor Biology and Bioactive Compounds, São Francisco University, Avenida São Francisco de Assis, 218, Jardim São José, Bragança Paulista 12916-900, SP, Brazil; cvcpalamim@gmail.com (C.V.C.P.); boschiero.matheus@gmail.com (M.N.B.); felipe.valencise@gmail.com (F.E.V.); 2Laboratory of Human and Medical Genetics, São Francisco University, Avenida São Francisco de Assis, 218, Jardim São José, Bragança Paulista 12916-900, SP, Brazil; 3Postgraduate Program in Health Science, São Francisco University, Avenida São Francisco de Assis, 218, Jardim São José, Bragança Paulista 12916-900, SP, Brazil

**Keywords:** COVID-19, case fatality rate, human development index, pandemic

## Abstract

The Human Development Index measures a region’s development and is a step for development debate beyond the traditional, economic perspective. It can also determine the success of a country’s response to the COVID-19 pandemic, mainly affecting the case fatality rate among severe cases of SARS-CoV-2 infection. We aimed to associate the Human Development Index with the case fatality rate due to COVID-19 in each Brazilian state and the Federal District, taking into account comorbidities and the need for invasive mechanical ventilation. We also evaluated the influence of the GINI index, number of intensive care unit beds, and occupied households in subnormal clusters on the case fatality rate. We performed an ecological study including two populations: COVID-19 individuals that did not require the mechanical ventilation protocol; and COVID-19 individuals under invasive mechanical ventilation. We performed a Pearson correlation test and a univariate linear regression analysis on the relationship between Human Development Index, Human Development Index—Education Level, Human Development Index—Life Expectancy, and Human Development Index—Gross National Income per capita and COVID-19 deaths. The same analyses were performed using the other markers. We grouped the patients with COVID-19 according to comorbidities and the need for invasive mechanical ventilation. Alpha = 0.05. We included 848,501 COVID-19 individuals, out of which 153,710 needed invasive mechanical ventilation and 314,164 died, and 280,533 COVID-19 individuals without comorbidity, out of which 33,312 needed invasive mechanical ventilation and 73,723 died. We observed a low negative Pearson correlation between the Human Development Index and death and a moderate negative Pearson correlation between the Human Development Index and deaths of individuals on invasive mechanical ventilation, with or without comorbidity. The univariate linear analysis showed the case fatality rate depends on at least 20–40% of the Human Development Index. In Brazil, regions with a low Human Development Index demonstrated a higher case fatality rate due to COVID-19, mainly in individuals who needed invasive mechanical ventilation, than regions with a higher Human Development Index. Although other indexes studied, such as intensive care unit beds and GINI, were also associated with the COVID-19 case fatality rate, they were not as relevant as the Human Development Index. Brazil is a vast territory comprising cultural, social, and economic diversity, which mirrors the diversity of the Human Development Index. Brazil is a model nation for the study of the Human Development Index’s influence on aspects of the COVID-19 pandemic, such as its impact on the case fatality rate.

## 1. Introduction

The coronavirus disease (COVID)-19 can spread through person-to-person contact and through the air, affecting people throughout whole countries. Individuals with comorbidities, such as diabetes mellitus, systemic arterial hypertension, and obesity, are more likely to develop a more severe course and progression of the disease [1,2,3]. In addition, factors such as social distancing compliance, personal hygiene methods, and governmental response to the pandemic seem to attenuate the impact of the COVID-19 [4,5]. In this context, markers such as the Human Development Index (HDI) can be valuable for measuring a region’s development. It can be considered a step for development debate beyond the traditional, economic perspective and can also determine the success of a country’s response to the COVID-19 pandemic, mainly affecting the case fatality rate among severe cases of Severe Acute Respiratory Syndrome Coronavirus 2 (SARS-CoV-2) infection [6,7].

Although one of the most important sociodemographic indexes, the HDI, solely, cannot deeply analyze a country’s response to COVID-19 due to its simplicity. Thus, jointly analyzing other socio, economic, and demographic indexes, such as the GINI index and the number of intensive care units (ICU) beds, which were already linked to COVID-19 in previous studies [8,9], is essential. Living conditions can also play a key role as a risk factor for COVID-19 since poor infrastructure and overcrowding, such as that observed in slums, can enhance SARS-CoV-2 dissemination [10,11]. Importantly, these indicators can help one to understand the dynamics of the COVID-19 pandemic and allow one to compare the general needs of certain areas, thus, allowing one to make decisions based on reliable indicators [12,13,14].

The HDI can be a valuable tool for better understanding the impact of the COVID-19 pandemic in certain areas and can also guide better public health measures since the factors used to calculate the HDI (life expectancy, education, and gross national income per capita) are associated with COVID-19 mortality and infection [15]. For instance, regions with high life expectancy perhaps suffer a higher impact from COVID-19 since older age is associated with the worst outcomes in COVID-19 [16]. Additionally, the higher the level of education, the more individuals are aware of the pandemic risks of not adhering to preventive measures, such as the use of masks or social distancing, thus, decreasing the spread of SARS-CoV-2 [15,17]. Finally, regions with higher gross national income per capita are perhaps more likely to have international travelers and can purchase more COVID-19 tests, which can enhance SARS-CoV-2 dissemination and decrease underreporting, respectively [15].

HDI scores can range from 0 (lower development; e.g., Niger (0.394), Central African Republic (0.397), and Chad (0.398) in 2019) to 1 (higher development; e.g., Norway (0.957), Ireland (0.955), and Switzerland (0.955) in 2019) [18,19]. Brazil, in 2020, was at position 84 worldwide, with an HDI of 0.765, behind other South American countries such as Argentina (0.845), Uruguay (0.817), and Chile (0.851) [19].

The literature described a correlation between SARS-CoV-2 infection and death prevalence due to COVID-19 and the HDI, as well as a country’s socioeconomic position [20,21,22], including two Brazilian studies performed in the Alagoas and Pernambuco states. These studies demonstrated higher mortality in regions with a low HDI score [23,24]. However, the data from these studies are conflicting. Regarding Brazil, there is a wide range of HDI scores among the Brazilian states and the Federal District (0.850 in the Federal District to 0.683 in the Alagoas state) [25]. To date, the main findings for the association between HDI and the case fatality rate in Brazil present only local perspectives alongside a scarce representation of Brazil’s national scenario.

Brazil’s vast territory explains the wide disparity among states and the Federal District in HDI and other markers, such as the GINI index and access to health support, which can compromise access to complex treatments, including mechanical ventilatory support. The Brazilian diversity can be a model to determine the influence of a country’s development on its ability to deal with a pandemic situation, such as COVID-19. In this context, we aimed to associate socio, economic, and demographic indexes, such as the HDI, ICU beds, the GINI index, and occupied households in subnormal clusters, with the case fatality rate due to COVID-19 in each Brazilian state and the Federal District taking into account comorbidities and the need for invasive mechanical ventilation (IMV).

## 2. Materials and Methods

We performed an ecological study using epidemiologic data (death due to COVID-19 and death due to COVID-19 among individuals who needed IMV) available at OpenDataSUS [26]. We computed the data from the Brazilian Ministry of Health according to the surveillance data of severe acute respiratory infection and data from the Information System platform for Epidemiological Surveillance of Influenza (SIVEP-Flu). The data were recorded during the first year of COVID-19 in Brazil after the first report in our country. In such a context, we retrieved the information for severe acute respiratory infection from December 2019 to April 2020 and for patients with COVID-19 (SARS-CoV-2 infection) from February 2020 to April 2021. Only four days of April’s first week were included in our study. Inclusion criteria: patients with a positive SARS-CoV-2 real-time polymerase chain reaction (RT-PCR) test and complete information on the need for ventilatory support, outcomes, and place of residence. Exclusion criteria: negative SARS-CoV-2 RT-PCR test or absence of classification of severe acute respiratory infection, absence of a description for the place of residence, or patients who lived in a country other than Brazil. We also excluded the patients who presented comorbidities or pregnancy in a second analysis. In addition, we excluded the individuals without gender data or outcomes information for both analyses. The complete flowchart of the patients included and excluded is shown in Figure 1.

We present the data as percentages calculated using the ratio between individuals who died due to COVID-19 and the sum of COVID-19 individuals who died or recovered in Brazilian states and the Federal District. Two analyses were performed for: (i) all COVID-19 individuals without mechanical ventilation protocol; and (ii) COVID-19 individuals who needed IMV. Additionally, for subgroups analysis, we considered the presence of confounders, such as comorbidities.

The HDI has three principles to classify a region as developed: life expectancy (HDI-LE); capacity to acquire knowledge, that is, mean and expected years of schooling (HDI-E); and access to resources for a decent standard of living, that is, gross national income per capita (HDI-GNI) [6,7,15,27]. We retrieved the HDI data of each Brazilian state and the Federal District from the AtlasBR [25], a private company licensed by the federal government. Unfortunately, the latest HDI score for each Brazilian state and the Federal District was from 2017. The Brazilian Institute of Geography and Statistics (IBGE) website (Instituto Brasileiro de Geografia e Estatística, in Portuguese) published the latest HDI in 2010 [28]. 

The GINI index and the number of occupied households in subnormal clusters were obtained from the IBGE website [29]. The GINI index measures the income inequality of a certain area. It varies from 0 to 1, with regions with values close to zero presenting lower inequality in contrast to regions with values close to 1, which present higher inequality [30]. Regarding the number of occupied households in subnormal clusters (slums, or favelas in Portuguese), the IBGE defines such clusters as “forms of irregular occupation of land owned by others for housing purpose, characterized by an irregular urban pattern, lack of essential public services and location in that have restrictions on occupancy” [29]. We retrieved the data from 2019 since it was the last update before the COVID-19 pandemic. We decided to present the absolute number of occupied households in subnormal clusters and the relative number (%) of occupied households in subnormal clusters, representing the proportion of this type of occupation and the total number of households [29].

Regarding ICU beds, the following markers were included in our study: the number of ICU beds in Brazil, ICU beds in the Brazilian public health system (SUS; in Portuguese, Sistema Único de Saúde), ICU beds in the Brazilian private health system, ICU beds per 10,000 inhabitants, ICU beds per 10,000 inhabitants in the Brazilian public health system, and ICU beds per 10,000 inhabitants in the Brazilian private health system (beneficiaries only). We retrieved the data from the Federal Council of Medicine, 2018 [9,31]. 

We performed a Pearson correlation test between HDI, HDI-E, HDI-LE, and HDI-GNI and the number of deaths due to COVID-19, considering the comorbidities and the need for IMV. The same statistical test was used to correlate the GINI index, the number/percentage of occupied households in subnormal clusters, and the number of ICU beds per 10,000 inhabitants (total, public health system, and private health system) with the number of deaths due to COVID-19, considering comorbidities and the need for IMV. We considered the following categorization for the Pearson correlation test: very high positive/negative correlation, 0.9 to 1.0; high positive/negative correlation, 0.7 to 0.9; moderate positive/negative correlation, 0.5 to 0.7; low positive/negative correlation, 0.30 to 0.50; and negligible correlation, 0.00 to 0.30. We present the correlation coefficient (CC) and the 95% confidence interval (95%CI) for the Pearson correlation test.

We performed a univariate regression analysis between HDI, GINI index, the number/percentage of occupied households in subnormal clusters, and the number of ICU beds per 10,000 inhabitants (total, public health system, and private health system) and deaths due to COVID-19. The analysis considered the presence of comorbidities and the need for IMV. For the univariate regression analysis, we described the R-squared (goodness-of-fit measure for linear regression models that indicates the percentage of the variance in the dependent variable that the independent variables explain collectively). This equation represents how an independent variable X (HDI, GINI index, the number/percentage of occupied households in subnormal clusters, and the number of ICU beds per 10,000 inhabitants (total, public health system, and private health system)) is related to a dependent variable Y (case fatality rate) and the 95% confidence bands of the best-fit line. In brief, the primary marker evaluated in our study was the HDI and the others were the GINI index, the number/percentage of occupied households in subnormal clusters, and the number of ICU beds per 10,000 inhabitants (total, public health system, and private health system).

We performed the statistical analysis using the Statistical Package for the Social Sciences software (IBM SPSS Statistics for Macintosh, Version 27.0) and the GraphPad Prism version 8.00 for Apple Mac, GraphPad Software, San Diego, CA, USA, www.graphpad.com (accessed on 21 March 2022). We used an alpha of 0.05 in all statistical analyses.

The data used in our study were publicly available. As the data did not contain personal data of patients, the study was consent-free since it does not present risks to the research participants.

## 3. Results

### 3.1. Inclusion of the COVID-19 Individuals and the HDI of the Brazilian States and the Federal District

We included 848,501 individuals with positive RT-PCR tests for SARS-CoV-2 in the first analysis from the original cohort. Additionally, we performed a second analysis in which we enrolled individuals with SARS-CoV-2 infection and with no comorbidities screened; the second analysis comprised 280,533 individuals. For the first and second analyses, respectively, 153,710 and 33,312 individuals were under IMV. We present the description of inclusion and exclusion for the participants and the distribution of ventilatory support and outcome (case fatality rate and clinical recovery) in Figure 1. 

We observed the highest HDI in the Federal District (0.850), followed by São Paulo (0.826) and Santa Catarina (0.808) states. In contrast, the lowest HDI occurred in Alagoas (0.683), Maranhão (0.687), and Piauí (0.697) states. Regarding the components of the HDI, we observed the highest HDI-E in São Paulo (0.828) state, followed by the Federal District (0.804) and Santa Catarina (0.779) state. The highest HDI-LE occurred in the Federal District (0.859), followed by São Paulo (0.796) and the Rio Grande do Sul (0.787) states. Finally, we observed the highest HDI-GNI in the Federal District (0.890), followed by Minas Gerais (0.875) and Santa Catarina (0.866) states (Figure 2 and Figure 3). 

The Brazilian state which accounted for highest overall case fatality rate was Sergipe (3643/5806; 62.7%) state, followed by Espírito Santo (4618/7668; 60.2%) and Roraima (1121/1957; 57.3%) states, whereas the state with highest death rate in individuals without comorbidities was Sergipe (1010/1847; 54.7%), followed by Espírito Santo (1146/2191; 52.3%) and Rondônia (1475/3177; 46.4%) states (Table 1; Figure 2). Regarding IMV, the states which presented a higher case fatality rate among all individuals were Roraima (702/747; 94.0%), Pará (3990/4357; 91.5%), and Paraíba (2950/3233; 91.2%) states. However, a higher difference between the case fatality rate of all individuals and individuals who needed IMV occurred in Mato Grosso (63.8%), Piauí (52.3%), and the Federal District (52.1%). Regarding the individuals who needed IMV and did not have a comorbidity, the highest case fatality rate occurred in Roraima (226/248; 91.1%), Alagoas (394/436; 90.3%), and Paraíba (585/661; 88.5%) states. In contrast, the higher difference between the case fatality rate of individuals without comorbidities and individuals without comorbidities who needed IMV occurred in Mato Grosso (64.3%), Alagoas (60.3%), and Piauí (58.9%) states (Table 1, Figure 3, and Appendix A). 

### 3.2. Association between Case Fatality Rate Due to COVID-19 and the HDI

We observed a negative correlation (*p*-value < 0.05) between HDI and case fatality rate (CC = −0.449; 95% CI = −0.708 to −0.083) and deaths without comorbidities (CC = −0.467; 95% CI = −0.720 to −0.106), between HDI-GNI and case fatality rate (CC = −0.397; 95% CI = −0.675 to −0.020) and deaths without comorbidities (CC = −0.446; 95% CI = −0.706 to −0.079), between HDI-E and case fatality rate (CC = −0.390; 95% CI = −0.670 to −0.011), and, finally, between HDI-LE and case fatality rate (CC = −0.444; 95% CI = −0.705 to −0.077) and deaths without comorbidities (CC = −0.481; 95% CI = −0.728 to −0.123) (Figure 2). 

### 3.3. Association between Case Fatality Rate Due to COVID-19 in Individuals Who Needed IMV and the HDI

We observed a negative correlation (*p*-value < 0.05) between HDI and case fatality rate (CC = −0.629; 95% CI = −0.815 to −0.328) and deaths without comorbidities (CC = −0.619; 95% CI = −0.809 to −0.313), between HDI-GNI and case fatality rate (CC = −0.618; 95% CI = −0.808 to −0.311) and deaths without comorbidities (CC = −0.654; 95% CI = −0.828 to −0.365), between HDI-E and case fatality rate (CC = −0.569; 95% CI = −0.780 to −0.241) and deaths without comorbidities (CC = −0.509; 95% CI = −0.745 to −0.160), and, finally, between HDI-LE and case fatality rate (CC = −0.569; 95% CI = −0.780 to −0.241) and deaths without comorbidities (CC = −0.582; 95% CI = −0.788 to −0.260) (Figure 3). In brief, all these correlation indexes were moderate.

### 3.4. Univariate Regression Analysis between the Case Fatality Rate Due to COVID-19 and the HDI

We included patients with COVID-19, considering comorbidity and the need for IMV (Figure 4). We described the best fit R^2^ for case fatality rate in individuals who needed IMV (R^2^ = 0.396), followed by individuals who needed IMV and had no comorbidities (R^2^ = 0.384); overall, for individuals without comorbidities R^2^ = 0.218 and, overall, for COVID-19 individuals R^2^ = 0.202. In addition, the equations for each analysis were: overall case fatality rate, case fatality rate = −103.7 × (HDI) + 119.2 (Figure 4A); overall case fatality rate in COVID-19 individuals without comorbidities, case fatality rate = −130.4 × (HDI) + 128.9 (Figure 4B); case fatality rate in COVID-19 individuals who received IMV, case fatality rate = −69.94 × (HDI) + 137.2 (Figure 4C); and case fatality rate in COVID-19 individuals who needed IMV and did not have any comorbidities, case fatality rate = −120.1 × (HDI) + 168.1 (Figure 4D). The case fatality rate was the dependent marker, and the HDI was the independent marker.

### 3.5. Association between the Case Fatality Rate Due to COVID-19 and Other Populational Features

In our data, we described three other populational features (GINI coefficient, ICU bed distribution, and the number of occupied households in subnormal clusters) which contribute to the case fatality rate diversity in Brazil. We present the features in Table 2, Appendix A.

We observed the highest GINI index of household income per capita, at average prices for the year, in the Federal District (0.583), followed by Pernambuco (0.578) and Acre (0.575) states, which represented the states with the most significant inequalities. In contrast, the lowest GINI indexes of household income per capita, at average prices for the year, occurred in Santa Catarina (0.429), Mato Grosso (0.457), and Góias (0.474) states (Table 2). Regarding the component of the GINI index of the average real monthly income of people aged 14 and over received in the reference month, for all jobs, at average prices for the year, we observed the highest index in Piauí (0.592) state, followed by Sergipe (0.560) state and the Federal District (0.553). Additionally, the lowest GINI indexes occurred in Santa Catarina (0.417), Mato Grosso (0.440), and Goiás (0.441) states (Table 2).

The Brazilian state which accounted for the highest number of total ICU beds per 10,000 inhabitants was Rio de Janeiro (3.79) state, followed by the Federal District (3.39) and São Paulo (2.72) state. In contrast, the state with the lowest number of total ICU beds per 10,000 inhabitants was Acre (0.90) state, followed by Roraima (0.92) and Amapá (1.03) states (Table 2). Additionally, the Brazilian state which accounted for the highest number of total ICU beds per 10,000 inhabitants in the public health system was Paraná (1.54) state, followed by the Rio Grande do Sul (1.33) and Minas Gerais (1.30) states, whereas the state with the lowest number of total ICU beds per 10,000 inhabitants in the public health system was Amapá (0.33) state, followed by Piauí (0.56) and Roraima (0.57) states (Table 2). Finally, the Brazilian state which accounted for the highest number of total ICU beds per 10,000 inhabitants in the private health system was Mato Grosso (10.63) state, followed by the Federal District (8.78) and Rio de Janeiro (8.70) state. In contrast, the state with the lowest number of total ICU beds per 10,000 inhabitants in the private health system was Santa Catarina (2.61) state, followed by the Minas Gerais (3.14) and the Rio Grande do Sul (3.31) states (Table 2).

Additionally, the Brazilian state which accounted for the highest number of occupied households in subnormal clusters (%) was Amazonas (34.59%) state, followed by the Espiríto Santo (26.10%) and Amapá (21.58%) states. In contrast, the state with the lowest number of occupied households in subnormal clusters (%) was Mato Grosso do Sul (0.74) state, followed by Santa Catarina (1.42) and Goiás (1.55) states (Table 2).

#### 3.5.1. Association between the Case Fatality Rate Due to COVID-19 and GINI Coefficient

We observed a positive correlation (*p*-value = 0.034) between the GINI of household income per capita, at average prices for the year, and the case fatality rate in patients without comorbidities who received IMV (CC = 0.409; 95% CI = 0.035 to 0.683) (Figure 5). The same index (GINI coefficient of household income per capita, at average prices for the year) had the best fit R^2^ for case fatality rate in individuals without comorbidities who needed IMV (R^2^ = 0.168) with the following equation: case fatality rate = 84.31 × (GINI) + 33.67 (Figure 6A). The other study groups did not present any significative correlation (Figure 5) or equation in the regression analysis (Appendix A).

#### 3.5.2. Association between the Case Fatality Rate Due to COVID-19 and ICU

We observed a negative correlation (*p*-value = 0.040) between the number of ICU beds in the public health system per 10,000 inhabitants and the case fatality rate in patients who received IMV (CC = −0.397; 95% CI = −0.675 to −0.020) (Figure 7). No significant correlation occurred between the case fatality rate (all patients, patients who received IMV, all patients without comorbidities, and patients without comorbidities who received IMV) and total number of ICU beds per 10,000 inhabitants and the ICU beds in the private health system per 10,000 inhabitants (Figure 7). In addition, we observed a significative best fit R^2^ for case fatality rate in individuals who needed IMV (R^2^ = 0.158) and the distribution of the ICU beds in the public health system per 10,000 inhabitants with the following equation: case fatality rate = −7.00 × (ICU beds in the public health system per 10,000 inhabitants) + 91.14 (Figure 6B). The other study groups did not present any significative equation in the regression analysis (Appendix A).

#### 3.5.3. Association between the Case Fatality Rate Due to COVID-19 and Occupied Households in Subnormal Clusters

The number of occupied households in subnormal clusters and the percentage of occupied households in subnormal clusters did not present any significative correlation (Appendix A) with the case fatality rate due to COVID-19 or any significative equation in the regression analysis (Appendix A) that could predict death due to COVID-19.

## 4. Discussion

The COVID-19 pandemic became the most urgent public health crisis worldwide [4] as it caused thousands of deaths daily. In Brazil, COVID-19 was responsible for more than 660,000 deaths and millions of confirmed cases and is increasing daily, despite vaccinations [32]. Due to its large dimensions, 8,510,345.538 km^2^, climate differences, and governmental public health management differences [33], the HDI is significantly different for each Brazilian state and the Federal District. The HDI index varies between Brazilian states and the Federal District, ranging from 0.683 for the Alagoas state to 0.850 for the Federal District and 0.826 for the São Paulo state [25]. Indirectly, the index is associated with different levels of impact from the disease in Brazil, as shown in our data for case fatality rate due to COVID-19, mainly in cases who needed IMV and required hospitalization support, which is directly associated with the development of each state and the Federal District.

Although we studied the COVID-19 pandemic from February 2020 to April 2021, reports showed at least two different waves during this period in Brazil. To face the pandemic and its first wave, the Brazilian federal government transferred around 30% more budget in the first quarters of 2020 to the states compared to in 2019; however, only 8% was spent [34]. Even though Brazil was able to increase the number of ICU beds per 10,000 and the number of health care workers in 2020 compared to 2019, it did not handle the COVID-19 pandemic well, resulting in some of the greatest numbers of confirmed cases and deaths in the world [34,35]. Studies showed the second wave was more severe than the first one, with more hospitalizations and deaths [36,37]. Furthermore, patients affected during the second wave appeared to suffer more severe symptoms since more younger individuals who needed invasive ventilatory support died and were more hypoxemic, and more invasive ventilatory support was also required [36,37]. Several factors might have contributed to these differences. For instance, during the first wave, the most prevalent strains of SARS-CoV-2 were B.1.1.28 and B.1.1.3. In contrast, the second wave included the P.1 and P.2 strains (or Gamma strains) first described in the city of Manaus in Amazonas. It coincided with an increase in hospitalizations [37,38].

The Gamma strains might be more transmissible than non-Gamma strains and more deadly, especially to younger people, with a higher need for hospitalizations, ICU, and mechanical ventilation [39,40,41]. In the case of Manaus city, where the health care system collapsed in the second wave due to a lack of oxygen cylinders, there were increased numbers of hospitalizations and deaths and several allegations of corruption [33], showing that Brazil was not prepared for the second wave and that the first wave drained most of the resources available.

In the study period, we observed a correlation between case fatality rate and HDI. Brazilian states and the Federal District with a lower HDI score were likely to present a higher case fatality rate in individuals with and without comorbidities and among individuals who needed IMV, compared to individuals in states and the Federal District with a higher HDI score, which is similar to the findings in previous studies [20,21,42]. The present study follows a new Brazilian study that evaluated 203 cities in the São Paulo state, encompassing nearly 93% of the state’s population, in which HDI-GNI and HDI-LE were associated with COVID-19 mortality [43].

In addition, an Italian study encompassing 20 Italian regions in an ecological study and a study containing 189 countries reported a positive correlation between HDI and the COVID-19 death rate [22,44]. Previous Brazilian studies reported the HDI to be significantly associated with cumulative COVID-19 cases and faster dissemination of the virus; that is, the higher the HDI, the more cases were reported and in a shorter period [8,45]. Interestingly, one Brazilian study, which evaluated the cities from the Ceará state, also observed a positive correlation between HDI and the incidence of COVID-19, which is maybe related to worse sanitary conditions and the fact that Fortaleza, the capital of Ceará state, is a hub that attracts tourism, mainly from Europe, which could, ultimately, have enhanced the dissemination of SARS-CoV-2 [14,46]. 

Additionally, the associated results from the more effective health care systems, responsible for identifying early, asymptomatic, and subclinical cases, stated that individuals who live in high-HDI regions are more prone to have chronic diseases, which enhanced the COVID-19 mortality rate, dissemination rate, and the number of total cases [22,44]. In addition, areas with enhanced HDI were associated with more vaccines, ICU beds, ventilators, physicians, and nurses [47], thus, attenuating the impact of the COVID-19 case fatality rate. Unfortunately, regions with low HDI scores tend to have worse access to the health system and diagnostic tools for COVID-19, resulting in underreporting of COVID-19 cases and deaths, such as in Brazil, especially in regions where neglected peoples (e.g., indigenous peoples) live [44,48,49,50,51]. Countries with low HDI scores also tend to have worse surveillance systems, allowing an enhanced number of underreports and less detection of asymptomatic and mildly symptomatic patients [15], which was also the case in Brazil [49].

Since one of the components of the HDI is life expectancy, the higher the HDI-LE, the older the people who live in the region. Since the elderly are more likely to experience to severe cases of COVID-19, mainly due to the senescent immune system, we expected a positive correlation between death and HDI-LE similar to the literature [44,52,53,54]. Unfortunately, in Brazil, the mortality rate of COVID-19 individuals in the public health system is more significant than in the private health system [33,55]. Thus, individuals who live in low-HDI-GNI regions have a higher mortality since the low purchasing power makes it difficult to buy a private health plan. Finally, the Brazilian regions with lower HDI, mainly in the north and northeast, have the lowest average number of years of study, from 18 to 29 years, which is a risk factor for death from COVID-19 [56,57]. Previous studies observed low educational attainment as a risk factor for death from COVID-19 [56,58]. Maybe, these individuals with a higher level of education have a better understanding of the disease, making this group seek early medical care, which increases the diagnoses rates [44]. Additionally, a previous study demonstrated a significant association between cumulative death and HDI (an increase of 0.1 in the HDI was associated with a nearly 40-fold increased likelihood of death) [22]. However, we observed that an increase in HDI of 0.1 was associated with a lower case fatality rate, ranging from −7% to −13%. In the same way, a study analyzing BRICS (Brazil, Russia, India, China, and South Africa) observed a negative correlation between HDI and new COVID-19 cases [59], which is also in accordance with our study, and, perhaps, it shows a similar pattern between the socioeconomic characteristics from these countries.

On the other hand, in Europe, the HDI was not associated with COVID-19 whatsoever. A study conducted in Barcelona, Spain, and another one that evaluated the whole European Union did not find an association between this socioeconomic index and COVID-19 cases and deaths, demonstrating the socioeconomic differences between Brazil and European countries [60,61]. From a global perspective, however, the countries with higher HDI scores reported more COVID-19 deaths [15], perhaps due to their better surveillance system, which may provide fewer underreports, and, since one of the components of HDI is life expectancy, countries with more older people had a higher mortality rate, as aforementioned.

Furthermore, our study also follows previously published Brazilian studies [23,24,45,62]. These studies observed a higher infection and case fatality rate in some specific, low-HDI areas of Brazil, such as Pernambuco and Alagoas states. Additionally, one of the most critical risks factors for COVID-19 is HDI [23,24,45,62]. Since Brazil is a vast territory with culture and socioeconomic characteristics that vary from region to region, it is expected for the Brazilian regions to have different HDI scores. However, regions with the lowest HDI score were the most affected by COVID-19, perhaps due to inadequate access to quality health care [63], leading to insufficient access to complex treatments such as IMV. Furthermore, high-HDI regions seemed to adhere more to non-pharmaceutical interventions, such as social isolation, hand hygiene, and facial masks, decreasing infection and case fatality rate [42].

Although the HDI in Rio de Janeiro is higher than in Amazonas, a higher mortality rate was observed in Rio de Janeiro, which contrasts with our results. Nevertheless, the Rio de Janeiro state has the second highest absolute number of subnormal clusters, just behind São Paulo state, which has poor structure and is overcrowding which might contribute to the COVID-19 spread [10,11] and, ultimately, to COVID-19 deaths. Furthermore, a recent report observed higher levels of COVID-19 death underreporting in the Amazonas state when compared to Rio de Janeiro [64]. These factors might explain, at least in part, the enhanced mortality in Rio de Janeiro, even though it has a higher HDI score.

We also observed a positive association between the GINI index and mortality due to COVID-19, which is similar to the current literature [65,66,67]. Interestingly, studies also reported an association between the GINI index and the incidence of COVID-19 cases [68,69]. Perhaps this association is because lower-income individuals tended to remain in jobs with a higher risk of contracting COVID-19, such as restaurants and hotels, which require high levels of person-to-person contact. Additionally, these individuals were usually unable to work from home [66,70]. Finally, we observed a correlation between ICU beds and COVID-19 mortality, especially between the public ICU beds, following previous Brazilian studies [9,71]. In Brazil, nearly 80% of the population depends on the public health system; however, even before the COVID-19 pandemic, almost 75% of health institutions reported insufficient numbers of ICU beds [72]. This might have played a crucial role in the collapse of the public health system in Brazil. For instance, in 2021, most Brazilian states reported a critical situation of occupancy in the ICU, with nearly 100% of the beds occupied [73]; there were also reports of patients being treated in hallways while awaiting ICU vacancy [74].

The number of COVID-19 infections and case fatality rate were more associated with patterns of socioeconomic vulnerability than population age or prior comorbidities. Brazil reported the first COVID-19 cases in São Paulo and Rio de Janeiro. After, the number of deaths rapidly increased in states with socioeconomic vulnerabilities, especially in northern and northeastern states, due to vulnerability, poor governmental response, and enhanced structural inequalities, which resulted in worse outcomes in the COVID-19 pandemic [75]. Although Brazil faces a challenge with the COVID-19 pandemic, more target responses are necessary through epidemiological data analysis, alongside social-economical indexes, such as HDI, and governmental investments.

Limitations: We retrieved the data regarding the COVID-19 cases and death from an online dataset attributed to the Ministry of Health; thus, the data input might have errors made by the person who typed the information. Loss of data and a limited number of markers to be evaluated were limitations of the dataset. Brazil reported many underreported COVID-19 cases, which could bias our analysis. We only analyzed the Brazilian states, making it difficult to extrapolate our data to the municipalities. The GINI index might be underestimated due to a lack of proper data; some states have relatively low access to ICUs, and the total subnormal cluster results might also be lacking data since some places in Brazil are difficult to access, which could bias our analysis using these indexes, thus, highlighting the importance of HDI.

## 5. Conclusions

In Brazil, regions with a low HDI and components demonstrated a higher case fatality rate due to COVID-19, mainly in individuals who needed IMV, than those with higher HDI. Although other indexes studied, such as number of ICU beds and GINI index, were also associated with the COVID-19 case fatality rate, they were not as relevant as the HDI. Brazil is a vast territory comprising cultural, social, and economic diversity, which mirrors the diversity of the HDI scores. Brazil is a model for the study of HDI influence on COVID-19 pandemic measures such as the COVID-19 case fatality rate.

## Figures and Tables

**Figure 1 ijerph-19-05306-f001:**
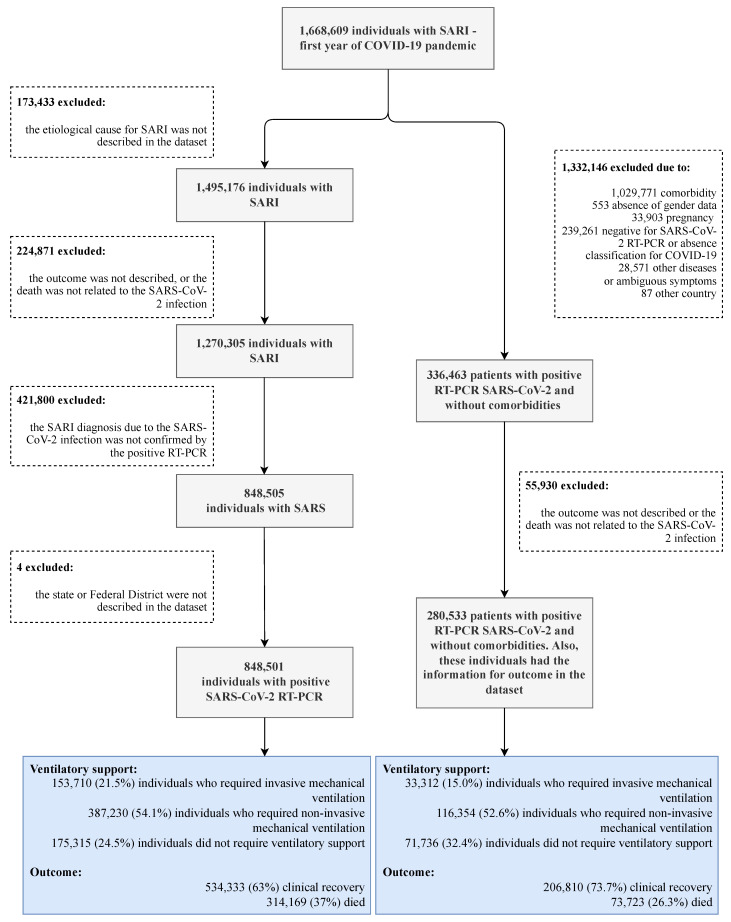
Description of inclusion and exclusion criteria for the study participants and the distribution of coronavirus disease COVID-19 individuals with ventilatory support and outcomes (death—case fatality rate and clinical recovery). SARI, severe acute respiratory infection; SARS, severe acute respiratory syndrome confirmed by a positive SARS-CoV-2 real-time polymerase chain reaction (RT-PCR) test. We obtained the data from OpenDataSUS [26].

**Figure 2 ijerph-19-05306-f002:**
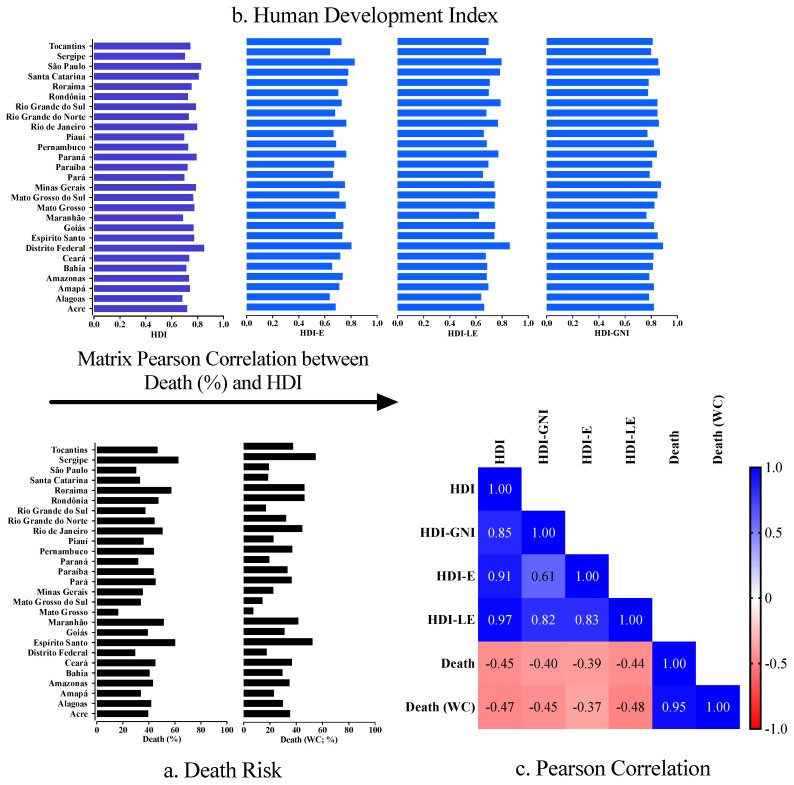
Distribution of the Human Development Index (HDI) (including the indicators for the educational level (HDI-E), life expectancy (HDI-LE), and gross national income per capita (HDI-GNI)) and the case fatality rate for coronavirus disease COVID-19 according to the Brazilian states and the Federal District. We presented the Pearson correlation matrix to compare HDI (HDI-LE, HDI-E, and HDI-GNI) with the case fatality rate for overall COVID-19 individuals and COVID-19 individuals without comorbidities. We considered the following categorization for the Pearson correlation test: very high positive/negative correlation, 0.9 to 1.0; high positive/negative correlation, 0.7 to 0.9; moderate positive/negative correlation, 0.5 to 0.7; low positive/negative correlation, 0.30 to 0.50; negligible correlation, 0.00 to 0.30. We presented an alpha error of 0.05 in all statistical analyses. We presented the case fatality rate as a percentage. WC, without comorbidities; %, percentage. We obtained the data from OpenDataSUS [26] and from the AtlasBR [25].

**Figure 3 ijerph-19-05306-f003:**
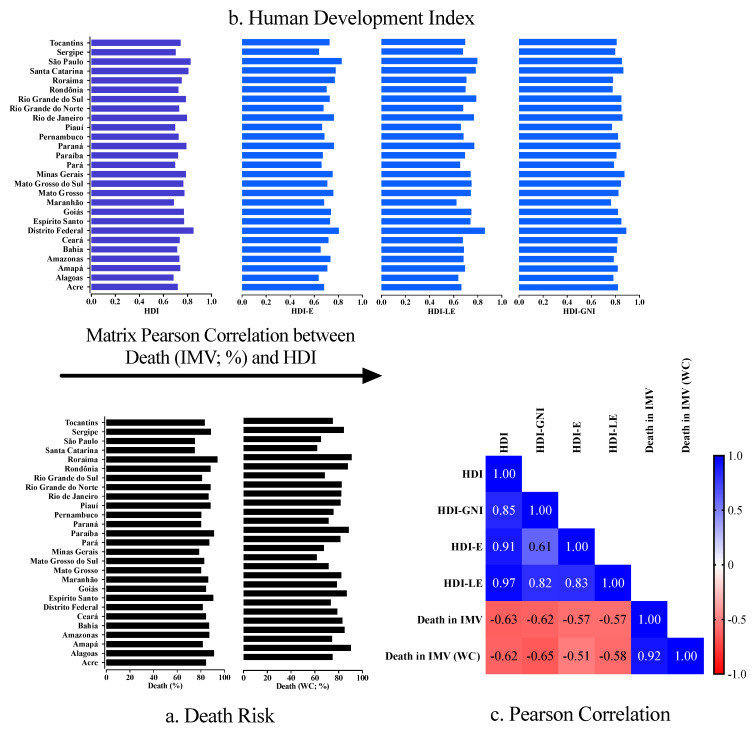
Distribution of the Human Development Index (HDI) (including the indicators for the educational level (HDI-E), life expectancy (HDI-LE), and gross national income per capita (HDI-GNI)) and the case fatality rate for coronavirus disease COVID-19 in individuals who needed invasive mechanical ventilation (IMV) according to the Brazilian states and the Federal District. We presented the Pearson correlation matrix to compare HDI (HDI-LE, HDI-E, and HDI-GNI) with the case fatality rate for overall COVID-19 individuals and COVID-19 individuals without comorbidities. We considered the following categorization for the Pearson correlation test: very high positive/negative correlation, 0.9 to 1.0; high positive/negative correlation, 0.7 to 0.9; moderate positive/negative correlation, 0.5 to 0.7; low positive/negative correlation, 0.30 to 0.50; negligible correlation, 0.00 to 0.30. We presented an alpha error of 0.05 in all statistical analyses. We presented the case fatality rate as a percentage. WC, without comorbidities; %, percentage. We obtained the data from OpenDataSUS [26] and from the AtlasBR [25].

**Figure 4 ijerph-19-05306-f004:**
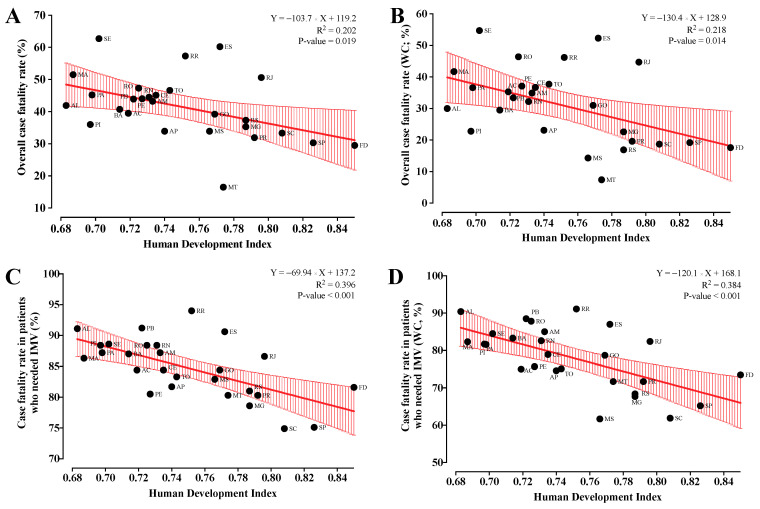
Univariate regression analysis between the case fatality rate due to coronavirus disease COVID-19 and the Human Development Index (HDI). (**A**) Overall case fatality rate. (**B**) The overall case fatality rate in COVID-19 individuals without comorbidities. (**C**) The case fatality rate in individuals who received invasive mechanical ventilation (IMV). (**D**) The case fatality rate in individuals who needed IMV and did not have comorbidities. The Y represents the case fatality rate as a dependent marker, and the Y describes the HDI as an independent marker. AC, Acre; AL, Alagoas; AP, Amapá; AM, Amazonas; BA, Bahia; CE, Ceará; ES, Espírito Santo; FD, Federal District; GO, Goiás; MA, Maranhão; MT, Mato Grosso; MS, Mato Grosso do Sul; MG, Minas Gerais; PA, Pará; PB, Paraíba; PR, Paraná; PE, Pernambuco; PI, Piauí; RJ, Rio de Janeiro; RN, Rio Grande do Norte; RS, Rio Grande do Sul; RO; Rondônia; RR, Roraima; SC, Santa Catarina; SP, São Paulo; SE, Sergipe; TO, Tocantins. We obtained the data from OpenDataSUS [26] and from the AtlasBR [25].

**Figure 5 ijerph-19-05306-f005:**
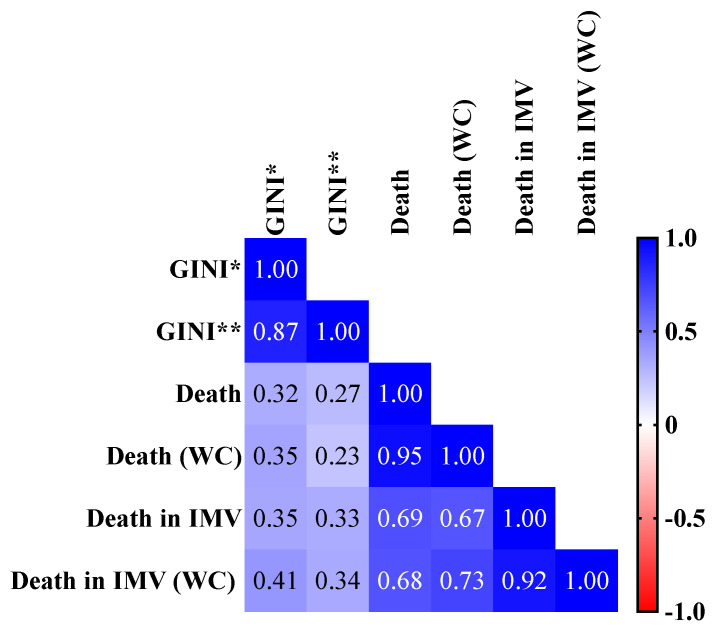
Pearson correlation matrix between GINI coefficient and case fatality rate according to the presence of comorbidities and the need for invasive mechanical ventilation (IMV). We presented the Pearson correlation matrix to compare the GINI coefficient (*, GINI of household income per capita, at average prices for the year; **, of the average real monthly income of people aged 14 and over actually received in the reference month, for all jobs, at average prices for the year) with the case fatality rate for overall coronavirus disease COVID-19 individuals and COVID-19 individuals without comorbidities (WC). We considered the following categorization for the Pearson correlation test: very high positive/negative correlation, 0.9 to 1.0; high positive/negative correlation, 0.7 to 0.9; moderate positive/negative correlation, 0.5 to 0.7; low positive/negative correlation, 0.30 to 0.50; negligible correlation, 0.00 to 0.30. We presented an alpha error of 0.05 in all statistical analyses. We presented the case fatality rate as a percentage. We obtained the data from OpenDataSUS [26] and from the Brazilian Institute of Geography and Statistics (IBGE) website (Instituto Brasileiro de Geografia e Estatística, in Portuguese) [29].

**Figure 6 ijerph-19-05306-f006:**
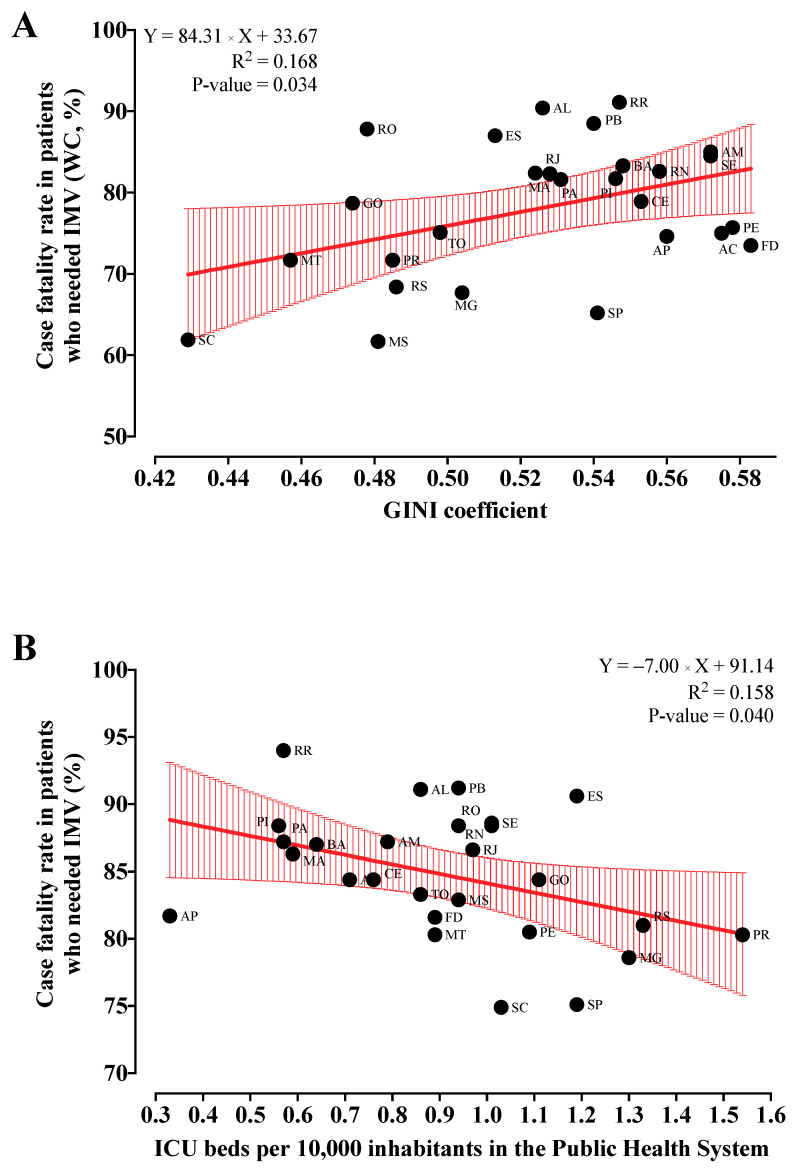
Univariate regression analysis between the case fatality rate due to coronavirus disease COVID-19 and the GINI coefficient and intensive care unit (ICU) beds per 10,000 inhabitants in the public health system. (**A**) The case fatality rate in individuals who needed IMV and did not have comorbidities. (**B**) The case fatality rate in individuals who received invasive mechanical ventilation (IMV). The Y represents the case fatality rate as a dependent marker, and the Y describes the GINI coefficient and ICU beds per 10,000 inhabitants at the public health system as an independent marker. AC, Acre; AL, Alagoas; AP, Amapá; AM, Amazonas; BA, Bahia; CE, Ceará; ES, Espírito Santo; FD, Federal District; GO, Goiás; MA, Maranhão; MT, Mato Grosso; MS, Mato Grosso do Sul; MG, Minas Gerais; PA, Pará; PB, Paraíba; PR, Paraná; PE, Pernambuco; PI, Piauí; RJ, Rio de Janeiro; RN, Rio Grande do Norte; RS, Rio Grande do Sul; RO; Rondônia; RR, Roraima; SC, Santa Catarina; SP, São Paulo; SE, Sergipe; TO, Tocantins. We obtained the data from OpenDataSUS [26], the Federal Council of Medicine website, Palamim and Marson (2020) [9,31], and from the Brazilian Institute of Geography and Statistics (IBGE) website (Instituto Brasileiro de Geografia e Estatística, in Portuguese) [29].

**Figure 7 ijerph-19-05306-f007:**
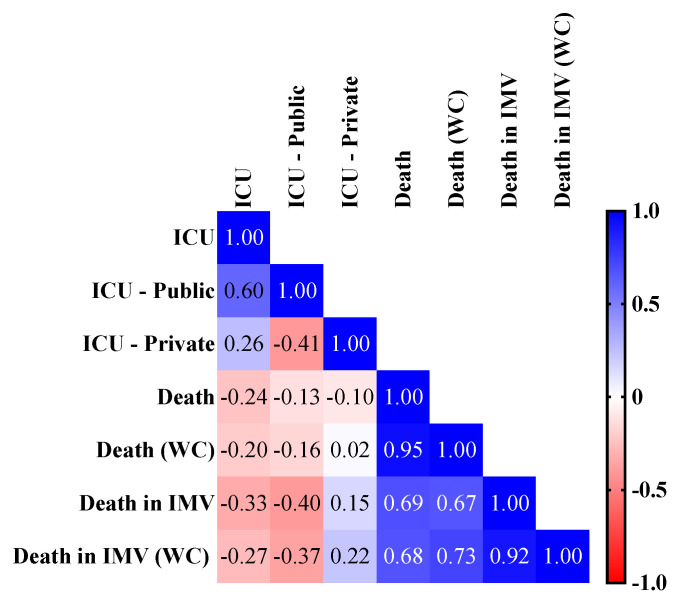
Pearson correlation matrix between intensive care unit (ICU) beds per 10,000 inhabitants (overall ICU beds per 10,000 inhabitants, overall ICU beds per 10,000 inhabitants in the public health system, and overall ICU beds per 10,000 inhabitants in the private health system) and case fatality rate according to the presence of comorbidities and the need for invasive mechanical ventilation (IMV). We presented the Pearson correlation matrix to compare the number of ICU beds per 10,000 inhabitants with the case fatality rate for overall COVID-19 individuals and COVID-19 individuals without comorbidities (WC). We considered the following categorization for the Pearson correlation test: very high positive/negative correlation, 0.9 to 1.0; high positive/negative correlation, 0.7 to 0.9; moderate positive/negative correlation, 0.5 to 0.7; low positive/negative correlation, 0.30 to 0.50; negligible correlation, 0.00 to 0.30. We presented an alpha error of 0.05 in all statistical analyses. We presented the case fatality rate as a percentage. We obtained the data from OpenDataSUS [26], the Federal Council of Medicine website, and Palamim and Marson (2020) [9,31].

**Table 1 ijerph-19-05306-t001:** Description of case fatality rate and cases of coronavirus disease COVID-19 in Brazil during the first year of the pandemic regarding the presence of comorbidities and the need for invasive mechanical ventilation (IMV) *.

States and the Federal District	With and without Comorbidities		Increase in Cases of Deaths %	Only without Comorbidities		Increase in Cases of Deaths %
	All Individuals -N (%)	IMV -N (%)		All Individuals -N (%)	IMV -N (%)	
Acre	929/2349 (39.5)	76/90 (84.4)	44.9	449/1273 (35.2)	24/32 (75.0)	39.8
Alagoas	3496/8353 (41.8)	1783/1958 (91.0)	49.2	793/2647 (30.0)	394/436 (90.3)	60.3
Amapá	1001/2956 (33.8)	628/769 (81.6)	47.8	290/1255 (23.1)	181/244 (74.1)	51.0
Amazonas	12,042/27,845 (43.2)	4142/4751 (87.1)	44.0	4571/13,098 (34.9)	1373/1616 (85.0)	50.1
Bahia	12,741/31,340 (40.6)	6367/7315 (87.0)	46.3	2385/8080 (29.5)	1030/1237 (83.2)	53.7
Ceará	14,884/33,020 (45.0)	5901/6991 (84.4)	39.3	4005/10,907 (36.7)	1334/1690 (78.9)	42.2
Federal District	6411/21,724 (29.5)	3556/4356 (81.6)	52.1	989/5628 (17.7)	516/702 (73.5)	55.8
Espírito Santo	4618/7668 (60.2)	1900/2098 (90.5)	30.3	1146/2191 (52.3)	400/460 (87.0)	34.7
Goiás	11,551/29,458 (39.2)	5341/6326 (84.4)	34.7	3422/11,036 (31.0)	1428/1814 (78.7)	47.7
Maranhão	4331/8402 (51.5)	955/1107 (86.2)	34.8	1403/3363 (41.7)	255/310 (82.2)	40.6
Mato Grosso	3172/19,226 (16.4)	992/1236 (80.2)	63.8	823/11,096 (7.4)	231/322 (71.7)	64.3
Mato Grosso do Sul	4527/13,351 (34.0)	2146/2590 (83.0)	49.0	619/4337 (14.3)	261/423 (61.7)	47.4
Minas Gerais	25,833/73,165 (35.3)	9061/11,535 (78.5)	43.2	4598/20,325 (22.6)	1461/2159 (67.7)	45.1
Pará	11,139/24,642 (45.2)	3990/4357 (91.5)	46.3	4057/11,082 (36.6)	1293/1584 (81.6)	45.0
Paraíba	5822/13,271 (43.8)	2950/3233 (91.2)	47.3	1240/3717 (33.3)	585/661 (88.5)	55.2
Paraná	15,619/48,978 (31.8)	7801/9719 (80.2)	48.4	3006/15,311 (19.6)	1495/2085 (71.7)	52.1
Pernambuco	11,759/26,749 (43.9)	1882/2337 (80.5)	36.5	4155/11,191 (37.1)	446/589 (76.1)	39.0
Piauí	3246/9021 (36.0)	1934/2188 (88.3)	52.3	633/2774 (22.8)	357/437 (81.7)	58.9
Rio de Janeiro	37,508/74,109 (50.6)	10,567/12,224 (86.4)	35.8	10,749/24,05 (44.6)	2282/2770 (82.3)	37.7
Rio Grande do Norte	3828/8605 (44.4)	1869/2115 (88.3)	43.9	643/1996 (32.2)	294/356 (82.5)	50.3
Rio Grande do Sul	20,753/55,587 (37.3)	10,813/13,349 (81.0)	43.7	2273/13,458 (16.9)	1152/1683 (68.4)	51.5
Rondônia	3523/7445 (47.3)	2268/2566 (88.3)	41.0	1475/3177 (46.4)	933/1063 (87.7)	41.3
Roraima	1121/1957 (57.2)	702/747 (94.0)	36.8	364/788 (46.2)	226/248 (91.1)	44.9
Santa Catarina	11,068/33,210 (33.3)	4770/6369 (74.9)	41.6	2097/11,200 (18.7)	869/1404 (61.9)	43.2
São Paulo	77,767/256,324 (30.3)	30,297/40,358 (75.0)	44.7	16,008/83,320 (19.2)	5471/8393 (65.2)	46.0
Sergipe	3643/5806 (62.7)	1835/2071 (88.6)	25.9	1010/1847 (54.7)	359/425 (84.5)	29.8
Tocantins	1836/3940 (46.6)	612/735 (84.5)	36.9	520/1379 (37.7)	127/169 (75.1)	37.4

*, we described the deaths/(deaths and recovery cases) data. We calculated the case fatality rate increase by the difference between the % of deaths in individuals who needed IMV and the % of deaths among all individuals. We obtained the data from OpenDataSUS [26]. N, number of individuals; %, percentage.

**Table 2 ijerph-19-05306-t002:** Description of GINI index *, intensive care unit (ICU) beds per 10,000 inhabitants **, and the number of occupied households in subnormal clusters *** in Brazil according to states and the Federal District.

States and Federal District	GINI ^a,b^		ICU Beds Per 10,000 Inhabitants			Number of Occupied Households in Subnormal Clusters (%) ^d^
			Total	Public Health System	Private Health System (Beneficiaries) ^c^	
Acre	0.575	0.528	0.90	0.71	3.54	19,148 (8.53)
Alagoas	0.526	0.480	1.45	0.86	5.18	64,568 (6.68)
Amapá	0.560	0.496	1.03	0.33	8.27	36,835 (21.58)
Amazonas	0.572	0.537	1.24	0.79	3.43	393,955 (34.59)
Bahia	0.548	0.537	1.32	0.64	6.49	469,677 (10.62)
Ceará	0.553	0.534	1.33	0.76	4.01	243,848 (9.20)
Federal District	0.583	0.553	3.39	0.89	8.78	62,179 (6.65)
Espírito Santo	0.513	0.468	2.72	1.19	5.60	306,439 (26.10)
Goiás	0.474	0.441	2.08	1.11	5.92	35,801 (1.55)
Maranhão	0.528	0.546	1.12	0.59	8.18	144,625 (7.85)
Mato Grosso	0.457	0.440	2.62	0.89	10.63	22,429 (1.99)
Mato Grosso do Sul	0.481	0.470	1.78	0.94	4.71	6766 (0.74)
Minas Gerais	0.504	0.486	2.06	1.30	3.14	231,385 (3.43)
Pará	0.531	0.527	1.18	0.57	6.42	432,518 (19.68)
Paraíba	0.540	0.538	1.51	0.94	5.49	64,225 (5.07)
Paraná	0.485	0.468	2.52	1.54	3.93	135,188 (3.57)
Pernambuco	0.578	0.551	1.96	1.09	6.31	327,090 (10.55)
Piauí	0.546	0.592	1.10	0.56	5.57	50,382 (5.49)
Rio de Janeiro	0.524	0.484	3.79	0.97	8.70	717,326 (12.63)
Rio Grande do Norte	0.558	0.549	1.71	0.94	5.20	41,868 (3.97)
Rio Grande do Sul	0.486	0.489	2.10	1.33	3.31	133,021 (3.50)
Rondônia	0.478	0.447	1.63	1.01	7.08	23,236 (4.37)
Roraima	0.547	0.532	0.92	0.57	6.29	3033 (2.12)
Santa Catarina	0.429	0.417	1.58	1.03	2.61	32,416 (1.46)
São Paulo	0.541	0.533	2.63	1.19	3.80	1066,813 (7.09)
Sergipe	0.572	0.560	1.48	1.01	3.47	53,203 (7.37)
Tocantins	0.498	0.475	1.43	0.86	8.37	9733 (2.14)

*, the GINI index was obtained from the Brazilian Institute of Geography and Statistics (IBGE) website (Instituto Brasileiro de Geografia e Estatística, in Portuguese) [29], and the calculation was performed in 2017; **, the number of occupied households in subnormal clusters was obtained from the IBGE website [29], and the calculation was performed in 2019; ***, the ICU beds distribution was obtained from the Federal Council of Medicine website and Palamim and Marson (2020) [9,31]. ^a^, GINI index of household income per capita, at average prices for the year (first column); ^b^, GINI index of the average real monthly income of people aged 14 and over actually received in the reference month, for all jobs, at average prices for the year (second column); ^c^, the proportion represents only the number of ICU beds among individuals that have access for the private health system (beneficiaries). In the public health system, we considered all Brazilian individuals; ^d^, we decided to present the absolute number of occupied households in subnormal clusters and the relative number (%) of occupied households in subnormal clusters, which represents the proportion of this type of occupation and the total number of households.

## Data Availability

Not applicable.

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
