# Peer review of "Human Development Index Is Associated with COVID-19 Case Fatality Rate in Brazil: An Ecological Study"

_ijerph, 2022, doi:10.3390/ijerph19095306_

Round 1
Reviewer 1 Report
Journal IJERPH Manuscript ID ijerph-1676203
Article : Title Human Development Index is Associated with COVID-19 Lethality in Brazil
REVIEW SUMMARY:
Although COVID-19 started in late 2019, its effects are still felt and millions of lives have been touched. The pandemic has revealed the lack of preparedness to cope with the threat of emerging viral pathogens, especially in developing countries. Brazil was one of the countries most severely affected by this pandemic. It may be helpful to examine socioeconomic indicators associated with COVID-19 mortality to assess potential readiness for future pandemics. Camila Palamim et al, have conducted a retrospective analysis of the Human Development Index (HDI) and COVID-19 mortality levels in Brazil. During this analysis, invasive mechanical ventilation and comorbidities were also taken into consideration. The statistical analysis of around 850 thousand samples from Brazil revealed a higher incidence of infection and mortality in low HDI states in Brazil.
Study of associations between COVID-19 mortality and HDI and other socioeconomic indicators has been conducted elsewhere and in Brazil, so the study is not novel. Furthermore, using HDI as the only socioeconomic indicator to correlate with COVID-19 mortality, especially for a country like Brazil that has a diverse ethnic/geographic/demographic composition, would be too simplistic. It was found that some states, like Rio de Janeiro, had relatively high HDI, but had significantly higher COVID-19 mortality rates compared to states like Amazonas with lower HDI, which calls for a much more comprehensive analysis taking into account other healthcare indices. I recommend the paper for major revision with analysis using additional healthcare indices apart from HDI.
MAJOR COMMENTS:
COMMENT 1: In this study, the authors did not mention the exact period during which the samples were taken for data analysis? They authors have only reported 1,668,609 cases in the first year of the pandemic. Could the authors clarify whether the first year refers to January 2020 to December 2020. Or whether the authors have taken the first 12 months after the first case of COVID-19 was reported in Brazil The first COVID-19 report in Brazil appears to have been around the end of February 2020).
COMMENT 2 : COVID-19 occured in Brazil in two waves in 2020- a first wave from April to August 2020 and a second wave from October to December 2020. According to a recent report, there were significant differences between the mortality and hospitalization rates for the two COVID-19 waves in Brazil in 2020 (Zeiser et al, 2022; The Lancet Regional Health - Americas 2022;6: 100107). The SARS-CoV-2- strains that were prevalent during these two COVID-19 waves must have been different. Aside from that, the health infrastructure would have been better prepared during the second wave than it was during the first wave, which struck without warning. It may also be possible that some healthcare facilities that were severely strained by the first wave were devastated by the second wave. The present study analyzed the year's data as a whole. The authors should reflect upon the rationale of analysis of the first year data as a whole, taking into account the above mentioned report on significant differences between the first two COVID-19 waves in Brazil in 2020.
COMMENT 3 : Several studies have shown that access to hospitalization and emergency care is a critical factor in COVID-19 mortality rates in developing countries. Does HDI reflect these differences in healthcare resource distribution? In a country like Brazil with disparities in the spatial distribution of healthcare resources due to factors such as geography and demography, adding other indices such as hospital beds or ICU beds per 100000 people could have provided a more comprehensive picture. I would appreciate if the authors were able to provide an additional table with HDI (%) ; hospital beds per 100000 people / ICU beds per 100000 people for the different states in Brazil, and reflect their findings.
COMMENT 4 : Rio de Janeiro has relatively high HDI compared to Amazonas, almost at par with São Paulo. But the COVID-19 mortality rate was much higher in Rio de Janeiro than Amazonas. It is expected the authors will comment on this anomaly, which is contrary to the paper's theme.
COMMENT 5 : Figure 2/3- Data as a Geospatial Color map - It would appreciated it if the authors could provide the data presented in Figure 2/3 for different states of Brazil in the form of a geospatial map that displays the three factors as separate colour gradient maps based on different values for the three indicators - 1) HDI 2) Matrix Spearman Correlation between Death (%) and HDI 3) Matrix Spearman Correlation between Death (IMV; %) and HDI). This will provide readers with a geospatial perspective of these three factors in various states/provinces of Brazil and shed light on any important geospatial patterns.
MINOR COMMENTS/SUGGESTIONS
- Figure numbering seems to be wrong – First figure (page 4) is labelled as Figure 2 ( line 152). Kindly correct this. There is another Figure 2 in page number 6 and legend for the figure on page 7- line 171).
- Figure 1- usage-‘individuals with SARS’: Also in the Figure 1 (Page 4)- ie ‘Description of inclusion and exclusion criteria for the study participants’ , it is mentioned ‘individuals with SARS’:It may be noted that SARS is used to refer to the Severe acute respiratory syndrome ( respiratory disease of zoonotic origin caused by severe acute respiratory syndrome coronavirus (SARS-CoV or SARS-CoV-1) - the 2003 outbreak. For the current COVID-19 pandemic – usage of 'COVID-19 for the disease and 'SARS-CoV-2' for the pathogen is recommended.
- FIGURE – 4-Regression Analysis between the lethality rate due to Coronavirus Disease (COVID)-19 and the Human Development Index (HDI):Please provide a key in the legends of the figure to the two letter abbreviations for various Brazilian states. Readers outside of Brazil could likely be unfamiliar with the two letter abbreviations used for states in Brazil.
Author Response
REVIEW SUMMARY:
Although COVID-19 started in late 2019, its effects are still felt and millions of lives have been touched. The pandemic has revealed the lack of preparedness to cope with the threat of emerging viral pathogens, especially in developing countries. Brazil was one of the countries most severely affected by this pandemic. It may be helpful to examine socioeconomic indicators associated with COVID-19 mortality to assess potential readiness for future pandemics. Camila Palamim et al, have conducted a retrospective analysis of the Human Development Index (HDI) and COVID-19 mortality levels in Brazil. During this analysis, invasive mechanical ventilation and comorbidities were also taken into consideration. The statistical analysis of around 850 thousand samples from Brazil revealed a higher incidence of infection and mortality in low HDI states in Brazil.
Study of associations between COVID-19 mortality and HDI and other socioeconomic indicators has been conducted elsewhere and in Brazil, so the study is not novel. Furthermore, using HDI as the only socioeconomic indicator to correlate with COVID-19 mortality, especially for a country like Brazil that has a diverse ethnic/geographic/demographic composition, would be too simplistic. It was found that some states, like Rio de Janeiro, had relatively high HDI, but had significantly higher COVID-19 mortality rates compared to states like Amazonas with lower HDI, which calls for a much more comprehensive analysis taking into account other healthcare indices. I recommend the paper for major revision with analysis using additional healthcare indices apart from HDI.
Reply: We thank the reviewer for the comments. We strongly agree that using only HDI would be too simplistic given Brazil is one of the largest and most culturally and socially countries in the world. Thus, we added other socio, economic and demographic indexes, such as the GINI index, number of occupied households in subnormal clusters, and number of the intensive care unit beds, to help understand the Brazilian scenario in face of the COVID-19 pandemic. In this context, we improved all manuscript sections and we included two supplementary materials. Thank you so much to support us with the great ideas and comments.
MAJOR COMMENTS:
COMMENT 1: In this study, the authors did not mention the exact period during which the samples were taken for data analysis? They authors have only reported 1,668,609 cases in the first year of the pandemic. Could the authors clarify whether the first year refers to January 2020 to December 2020. Or whether the authors have taken the first 12 months after the first case of COVID-19 was reported in Brazil The first COVID-19 report in Brazil appears to have been around the end of February 2020).
Reply: The authors appreciate the reviewer comment. We did not inform the period of data collection in the first version of the manuscript. To improve the readability, we informed that the patients’ data were imputed during the first year of COVID-19 in Brazil after the first report in our country. In such context, the data retrieved the information for Severe Acute Respiratory Infection from December 2019 to April 2020; and for patients with COVID-19 (SARS-CoV-2 infection) from February 2020 to April 2021. In the first week of April, only four days were input. We included the information in the methods session.
COMMENT 2: COVID-19 occured in Brazil in two waves in 2020- a first wave from April to August 2020 and a second wave from October to December 2020. According to a recent report, there were significant differences between the mortality and hospitalization rates for the two COVID-19 waves in Brazil in 2020 (Zeiser et al, 2022; The Lancet Regional Health - Americas 2022;6: 100107). The SARS-CoV-2- strains that were prevalent during these two COVID-19 waves must have been different. Aside from that, the health infrastructure would have been better prepared during the second wave than it was during the first wave, which struck without warning. It may also be possible that some healthcare facilities that were severely strained by the first wave were devastated by the second wave. The present study analyzed the year's data as a whole. The authors should reflect upon the rationale of analysis of the first year data as a whole, taking into account the above mentioned report on significant differences between the first two COVID-19 waves in Brazil in 2020.
Reply: We thank the reviewer for the comments. We added the following excerpt addressing both the investment in health and the first and second waves of COVID-19 in Brazil:
“Although we have studied the COVID-19 pandemic from February 2020 to April 2021, reports showed at least two different waves during this period in Brazil. To face the pandemic and its first wave, the Brazilian federal government transferred around 30% more budget in the first quarters of 2020, compared to 2019, to the states; however, only 8% was spent [34]. Even though Brazil was able to increase the amount of ICU beds per 10,000 and health care workers in 2020 compared to 2019, it did not handle well the COVID-19 pandemic, accounting for one of the greatest numbers of confirmed cases and deaths worldwide [34,35]. Studies have shown the second wave was more severe than the first one, with more hospitalizations and deaths [36,37]. Furthermore, patients affected during the second wave appeared to be more severe since younger individuals who needed invasive ventilatory support died, more hypoxemic and more invasive ventilatory support was also required [36,37]. Several factors might have contributed to these differences. For instance, during the first wave, the most prevalent strains of SARS-CoV-2 were B.1.1.28 and B.1.1.3. In contrast, the second wave accounted for the P.1 and P.2 strains (or Gamma strains), first described in the city of Manaus in Amazonas. It coincided with an increase in hospitalizations [37,38].
The Gamma strains might be more transmissible than non-Gamma strains and more deadly, especially to the youngest, with a higher need for hospitalizations, ICU, and mechanical ventilation rate [39–41]. The case of Manaus city, in which the healthcare system collapsed in the second wave, with a lack of oxygen cylinders, enhanced hospitalizations and deaths, and several allegations of corruption [33] shows that Brazil was not prepared for the second wave and how the first wave drained most of the resources available.”
COMMENT 3: Several studies have shown that access to hospitalization and emergency care is a critical factor in COVID-19 mortality rates in developing countries. Does HDI reflect these differences in healthcare resource distribution? In a country like Brazil with disparities in the spatial distribution of healthcare resources due to factors such as geography and demography, adding other indices such as hospital beds or ICU beds per 100000 people could have provided a more comprehensive picture. I would appreciate if the authors were able to provide an additional table with HDI (%) ; hospital beds per 100000 people / ICU beds per 100000 people for the different states in Brazil, and reflect their findings.
Reply: We thank the reviewer for the comments and the opportunity to improve our study. We added the ICU beds analysis, as requested and we also used other features, such as GINI index and the number of occupied households in subnormal clusters. We added the information in all manuscript sections.
COMMENT 4: Rio de Janeiro has relatively high HDI compared to Amazonas, almost at par with São Paulo. But the COVID-19 mortality rate was much higher in Rio de Janeiro than Amazonas. It is expected the authors will comment on this anomaly, which is contrary to the paper's theme.
Reply: We thank the reviewer for the comment. We agree it is surprising a higher mortality rate in the Rio de Janeiro state, even though it has a higher HDI, which contrast, with our results, thus, we added the following excerpt in order to explain, at least in part, this anomaly:
“Although the HDI in Rio de Janeiro was higher than in Amazonas, a higher mortality rate was observed in Rio de Janeiro, which contrasts with our results. Nevertheless, the Rio de Janeiro state is the second one with the highest absolute number of subnormal clusters, just behind São Paulo state, which has poor structure and is over-crowding, which might contribute to the COVID-19 spread [10,11] and ultimately do COVID-19 death. Furthermore, a recent report observed a higher COVID-19 death underreporting in the Amazonas state when compared to Rio de Janeiro [64]. These factors might explain, at least in part, the enhanced mortality in Rio de Janeiro, even with higher HDI.”
COMMENT 5: Figure 2/3- Data as a Geospatial Color map - It would appreciated it if the authors could provide the data presented in Figure 2/3 for different states of Brazil in the form of a geospatial map that displays the three factors as separate colour gradient maps based on different values for the three indicators - 1) HDI 2) Matrix Spearman Correlation between Death (%) and HDI 3) Matrix Spearman Correlation between Death (IMV; %) and HDI). This will provide readers with a geospatial perspective of these three factors in various states/provinces of Brazil and shed light on any important geospatial patterns.
Reply: The authors thank the reviewer for the important contribution. The authors decided to include the geospatial color map for all features. In this context, we included the maps as Supplementary Material 2. The figure had high resolution and we decided to submit it as Supplementary Material to facilitate its visualization.
MINOR COMMENTS/SUGGESTIONS
- Figure numbering seems to be wrong – First figure (page 4) is labelled as Figure 2 ( line 152). Kindly correct this. There is another Figure 2 in page number 6 and legend for the figure on page 7- line 171).
Reply: We corrected the numbers.
- Figure 1- usage-‘individuals with SARS’: Also in the Figure 1 (Page 4)- ie ‘Description of inclusion and exclusion criteria for the study participants’ , it is mentioned ‘individuals with SARS’:It may be noted that SARS is used to refer to the Severe acute respiratory syndrome ( respiratory disease of zoonotic origin caused by severe acute respiratory syndrome coronavirus (SARS-CoV or SARS-CoV-1) - the 2003 outbreak. For the current COVID-19 pandemic – usage of 'COVID-19 for the disease and 'SARS-CoV-2' for the pathogen is recommended.
Reply: We corrected Figure 1. The corrections were based on reviewers 1 and 3 comments. Thank you to provide an important contribution to our study.
- FIGURE – 4-Regression Analysis between the lethality rate due to Coronavirus Disease (COVID)-19 and the Human Development Index (HDI): Please provide a key in the legends of the figure to the two letter abbreviations for various Brazilian states. Readers outside of Brazil could likely be unfamiliar with the two letter abbreviations used for states in Brazil.
Reply: The authors thank the reviewer and we included the abbreviations:
“AC, Acre; AL, Alagoas; AP, Amapá; AM, Amazonas; BA, Bahia; CE, Ceará; ES, Espírito Santo; FD, Federal District; GO, Goiás; MA, Maranhão; MT, Mato Grosso; MS, Mato Grosso do Sul; MG, Minas Gerais; PA, Pará; PB, Paraíba; PR, Paraná; PE, Pernambuco; PI, Piauí; RJ, Rio de Janeiro; RN, Rio Grande do Norte; RS, Rio Grande do Sul; RO; Rondônia; RR, Roraima; SC, Santa Catarina; SP, São Paulo; SE, Sergipe; TO, Tocantins.”

Reviewer 2 Report
I appreciate the data used, but there are some observations I would like you to take into consideration:
- I suggest to organize the abstract, to the normal structure, without any abbreviations, without mentioning numbers for the results, add conclusions;
- - for the introduction section: when introducing HDI as a powerful indicator, that practically centers your paper, I expect to offer much more information that can explain your motivation, if not, it appears as if you have just took the idea from other countries and extend the plan for Brazil;
- for rows 50, 51 I suggest to add references, there are some affirmations that need to be supported with data from the literature;
- - for the section Materials and method: no need for so many details regarding the statistical formulas. Further more, please add in this section the the inclusion+ exclusion criteria, and explain if you have asked and obtained the approval for study design and data used
- please offer the motivation for the two analysis used
- for the section results, paragraph between rows 127-134, are not the own results, but results from data from the Minister of Health, please do not include them in your results, but in the discussion section.
- for the discussion section- we have also noticed that all correlations are moderate, so maybe there are also other indicators you could link to the lethality rate
- I like and think that figure 2 is very well illustrating the patients enrolment
Author Response
Comments and Suggestions for Authors
I appreciate the data used, but there are some observations I would like you to take into consideration:
- I suggest to organize the abstract, to the normal structure, without any abbreviations, without mentioning numbers for the results, add conclusions;
Reply: The authors thank the reviewer, and we included some corrections in the manuscript. We removed the abbreviations, and we improve the conclusions.
- - for the introduction section: when introducing HDI as a powerful indicator, that practically centers your paper, I expect to offer much more information that can explain your motivation, if not, it appears as if you have just took the idea from other countries and extend the plan for Brazil;
Reply: We thank the reviewer for the comment. We have added some explanation on why HDI is a useful index to evaluate the COVID-19 mortality rate, as follows:
“The HDI has three principles to classify a region as developed; life expectancy (HDI-LE), capacity to acquire knowledge, that is, mean and expected years of schooling (HDI-E), and access to resources for a decent standard of living, that is, gross national income per capita (HDI-GNI) [6,7,15,27]. We retrieved the HDI data of each Brazilian state and Federal District from the AtlasBR [25], a private company licensed by the Federal Government. Unfortunately, the latest HDI from each Brazilian state and Federal District was from 2017. The Brazilian Institute of Geography and Statistics (IBGE) website (Instituto Brasileiro de Geografia e Estatística, in Portuguese) published the latest HDI in 2010 [28].
The GINI index and the number of occupied households in subnormal clusters were obtained on the IBGE website [29]. The GINI index measures the income inequality of a certain area. It varies from 0 to 1, in which regions with values close to zero present lower inequality, in contrast with regions with values close to 1, which present higher inequality [30]. Regarding the number of occupied households in subnormal clusters (slums, or Favelas in Portuguese), the IBGE defines it as “forms of irregular occupation of land owned by others for housing purpose, characterized by an irregular urban pattern, lack of essential public services and location in that have restrictions on occupancy” [29]. We retrieved the data from 2019 since it was the last update before the COVID-19 pandemic. We decided to present the absolute number of occupied households in subnormal clusters and the relative number (%) of occupied households in subnormal clusters, representing the proportion of this type of occupation and the total number of households [29].
Regarding ICU beds, the following markers were included in our study: the number of ICU beds in Brazil, ICU beds in the Brazilian Public Health System (SUS, in Portuguese, Sistema Único de Saúde), ICU beds in the Brazilian Private Health System, ICU beds per 10,000 inhabitants, ICU beds per 10,000 inhabitants in the Brazilian Public Health System, and ICU beds per 10,000 inhabitants in the Brazilian Private Health System (beneficiaries only). We retrieved the data from the Federal Council of Medicine, 2018 [9,31].”.
- for rows 50, 51 I suggest to add references, there are some affirmations that need to be supported with data from the literature;
Reply: Now, we had references for the excerpt, and we corrected the paragraph based on the reviewers' reports.
- - for the section Materials and method: no need for so many details regarding the statistical formulas. Further more, please add in this section the inclusion+ exclusion criteria, and explain if you have asked and obtained the approval for study design and data used
Reply: Dear reviewer we deleted some parts of the methods section. We detailed the inclusion criteria:
“Inclusion criteria: patients with positive SARS-CoV-2 real time-polymerase chain reaction (RT-PCR) and complete information on the need for ventilatory support, outcomes, and place of residence. Exclusion criteria: negative SARS-CoV-2 RT-PCR or absence of classification of the severe acute respiratory infection, absence of a description for the place of residence, or patients who live in a country other than Brazil. We also excluded the patients who presented comorbidities OR pregnancy in a second analysis.”.
We also included the following excerpt about the study approval:
“The data used in our study were made publicly available, not containing personal data of patients being the study was consent-free since it is not presenting risks to the research participants.”
- please offer the motivation for the two analysis used
Reply: The authors thank the reviewer. We did the correlation to represent the linear relationship between two variables, mainly HDI and the case fatality rate due to SARS-CoV-2 in Brazil. On the other hand, we also calculate the regression to determine the best formula (line) to represent the association between both variables.
- for the section results, paragraph between rows 127-134, are not the own results, but results from data from the Minister of Health, please do not include them in your results, but in the discussion section.
Reply: Dear reviewer, we performed several corrections in the text, and we removed some excerpts. Also, another reviewer suggested keeping this data and including a geographical map. To respond to both reviewers, we decided to keep the text with some corrections and to include the new figure as supplementary material.
- for the discussion section- we have also noticed that all correlations are moderate, so maybe there are also other indicators you could link to the lethality rate
Reply: We thank the reviewers' comments. We added several other indicators, such as the GINI index, the number of occupied households in subnormal clusters, and the number of intensive care unit beds, to help understand the Brazilian scenario in face of the COVID-19 pandemic.
- I like and think that figure 2 is very well illustrating the patients enrolment
Reply: The authors thank the reviewer. Also, the figure 2 was placed as one.

Reviewer 3 Report
While there were numerous grammar issues in the paper, the analysis was sound, the conclusions did not overreach, and the background and discussion put the findings into context effectively.
Throughout-consider using mortality rate or fatality rate instead of lethality rate.
Introduction
“The HDI used the 51 adult literacy and combined enrollment ratios for the ability to acquire knowledge (HDI- 52 E); life expectancy for the healthy life-principle (HDI-LE); and adjusted Gross Domestic 53 Product (GDP) for the standard of living (HDI-GNI).”
I do not fully understand what you mean by the ability to acquire knowledge; please clarify. Also, please clarify the healthy-life principle.
Methods
“We 83 computed the data from the Brazilian Ministry of Health according to the surveillance 84 data of SARS and from the Information System platform for Epidemiological Surveillance 85 of Influenza (SIVEP-Flu).”
I think SARS should be SARAS-CoV2 here
Figure 1
“the outcome was not described or the death was related with the SARS-CoV2 infection”
should this say the death was NOT related to CoV2?
“the SARS diagnosis was related with other etiologic factor”
I would like a better explanation of this.
Results
“Y=-103.7*X + 210 119.2 (Figure 4A); (overall lethality rate in COVID-19 individuals without comorbidities) 211 Y=-130.4*X + 128.9 (Figure 4B); (lethality rate in individuals who received IMV) Y=-69.94*X 212 + 137.2 (Figure 4C); and (lethality rate in individuals who needed IMV and did not have 213 any comorbidities) Y=-120.1*X + 168.1 (Figure 4D)”
I would replace the theoretical (Y and X) with the actual names; ie mortality rate = -103.7*HDI+210, etc. This will help tie a lengthy set of equations to the nice figures below.
Discussion
“The Brazilian HDI was positively asso- 240 ciated with lethality rate and cumulative COVID-19 cases and faster dissemination of the 241 virus; that is, the higher the HDI, the more cases are reported and in a shorter period 242 [23,24]”
I disagree with this statement. In this study, you demonstrate a negative association of HDI and death. In your reference 23 and 24, HDI is indeed associated with higher case numbers, but I see no mention of an association with fatality rates. In short, in none of your manuscript or these references do I see a positive association between HDI and lethality rate and in yours I see evidence for a negative association.
Author Response
Comments and Suggestions for Authors
While there were numerous grammar issues in the paper, the analysis was sound, the conclusions did not overreach, and the background and discussion put the findings into context effectively.
Reply: We thank the reviewer for the comments. We have corrected the grammar issues addressed.
Throughout-consider using mortality rate or fatality rate instead of lethality rate.
Reply: The authors thank the reviewer for the important contribution. We corrected the text, and we used the case fatality rate in the new version of the manuscript.
Introduction
“The HDI used the 51 adult literacy and combined enrollment ratios for the ability to acquire knowledge (HDI- 52 E); life expectancy for the healthy life-principle (HDI-LE); and adjusted Gross Domestic 53 Product (GDP) for the standard of living (HDI-GNI).”
I do not fully understand what you mean by the ability to acquire knowledge; please clarify. Also, please clarify the healthy-life principle.
Reply: The authors agree the construction of the sentence was confusing, thus we rewrote it to make it clearer, as follows:
“The HDI has three principles to classify a region as developed; life expectancy (HDI-LE), capacity to acquire knowledge, that is, mean and expected years of schooling (HDI-E), and access to resources for a decent standard of living, that is, gross national income per capita (HDI-GNI) [6,7,15,27]. We retrieved the HDI data of each Brazilian state and Federal District from the AtlasBR [25], a private company licensed by the Federal Government. Unfortunately, the latest HDI from each Brazilian state and Federal District was from 2017. The Brazilian Institute of Geography and Statistics (IBGE) website (Instituto Brasileiro de Geografia e Estatística, in Portuguese) published the latest HDI in 2010 [28].”
Methods
“We 83 computed the data from the Brazilian Ministry of Health according to the surveillance 84 data of SARS and from the Information System platform for Epidemiological Surveillance 85 of Influenza (SIVEP-Flu).”
I think SARS should be SARAS-CoV2 here
Reply: We thank the reviewer for the comments. The term SARS is correct and according to the Brazilian dataset (website). The dataset comprised SARS due to SARS-CoV-2 and other Severe Acute Respiratory Infections. However, in our study, only individuals with SARS due to SARS-CoV-2 were included. To remove the doubts and to pair the information presented in the text with the Figure 1, we corrected the excerpt as follows:
“We computed the data from the Brazilian Ministry of Health according to the surveillance data of Severe Acute Respiratory Infection and from the Information System platform for Epidemiological Surveillance of Influenza (SIVEP-Flu).”
Figure 1
“the outcome was not described or the death was related with the SARS-CoV2 infection”
should this say the death was NOT related to CoV2?
Reply: We corrected the information. Now, we have the following description: “224,871 excluded: the outcome was not described, or the death was not related to the SARS-CoV-2 infection”.
“the SARS diagnosis was related with other etiologic factor”
I would like a better explanation of this.
Reply: The authors thank the reviewer. We include minor corrections in Figure 1. The Brazilian Ministry of Health included all cases of patients with clinical symptoms like SARS symptoms in the same dataset. In this way, the patients are classified as COVID-19, other virus infection, other etiologic agents, or no determined etiological agent. To present clear information, we included the abbreviation of SARI (severe acute respiratory infection) for all patients and the abbreviation of SARS for patients if positive RT-PCR. The information is presented in Figure 1.
The present description was included in Figure 1: “421,800 excluded: the SARI diagnosis due to SARS-CoV-2 was not confirmed by the positive RT-PCR.”
Results
“Y=-103.7*X + 210 119.2 (Figure 4A); (overall lethality rate in COVID-19 individuals without comorbidities) 211 Y=-130.4*X + 128.9 (Figure 4B); (lethality rate in individuals who received IMV) Y=-69.94*X 212 + 137.2 (Figure 4C); and (lethality rate in individuals who needed IMV and did not have 213 any comorbidities) Y=-120.1*X + 168.1 (Figure 4D)”
I would replace the theoretical (Y and X) with the actual names; ie mortality rate = -103.7*HDI+210, etc. This will help tie a lengthy set of equations to the nice figures below.
Reply: We corrected the sentence as suggested by the reviewer:
“(overall case fatality rate) case fatality rate = -103.7*(HDI) + 119.2 (Figure 4A); (overall case fatality rate in COVID-19 individuals without comorbidities) case fatality rate = -130.4*(HDI) + 128.9 (Figure 4B); (case fatality rate in individuals who received IMV) case fatality rate = -69.94*(HDI) + 137.2 (Figure 4C); and (case fatality rate in individuals who needed IMV and did not have any comorbidities) case fatality rate = -120.1*(HDI) + 168.1 (Figure 4D). The case fatality rate is the dependent marker and the HDI is the independent marker.”
Discussion
“The Brazilian HDI was positively asso- 240 ciated with lethality rate and cumulative COVID-19 cases and faster dissemination of the 241 virus; that is, the higher the HDI, the more cases are reported and in a shorter period 242 [23,24]”
I disagree with this statement. In this study, you demonstrate a negative association of HDI and death. In your reference 23 and 24, HDI is indeed associated with higher case numbers, but I see no mention of an association with fatality rates. In short, in none of your manuscript or these references do I see a positive association between HDI and lethality rate and in yours I see evidence for a negative association.
Reply: We thank the review. We agree with the reviewer. We misused the word positively, thus we have changed it to “significantly” as follows:
“Previous Brazilian studies reported the HDI to be significantly associated with cumulative COVID-19 cases and faster dissemination of the virus; that is, the higher the HDI, the more cases are reported and in a shorter period [8,45].”

Round 2
Reviewer 1 Report
The authors have made substantial revisions to the manuscript and incorporated my suggestions in the revised version. Apart from HDI, the authors have included other socio-economic/healthcare indices like GINI index which measures economic inequality in a population, number of ICU beds, etc as part of the analysis. The period of data collection was also updated in the revised materials & methods. In the updated discussion the authors have addressed the various factors that caused the difference in disease severity in the two COVID-19 waves that hit Brazil in 2020. Authors have also discussed factors like overcrowding and poor structure (infrastructure ?) in Rio de Janeiro which may have contributed to higher mortality despite relatively high HDI. I appreciate the authors for coming up with a detailed geospatial map showing fatality rates vs other indices in a short time. Also, minor corrections like Correction of the figure numbers, updating the pathogen name to SARS-CoV-2 in Figure 1 and providing a key for the two-letter abbreviations used for states in Brazil
I recommend the revised manuscript for publication.